# Nonalcoholic Fatty Liver Disease Induced by High-Fat Diet in C57bl/6 Models

**DOI:** 10.3390/nu11123067

**Published:** 2019-12-16

**Authors:** Leonardo Recena Aydos, Luane Aparecida do Amaral, Roberta Serafim de Souza, Ana Cristina Jacobowski, Elisvânia Freitas dos Santos, Maria Lígia Rodrigues Macedo

**Affiliations:** 1Postgraduate Program in Health and Development in the Midwest Region, Medical School, Federal University of Mato Grosso do Sul, Campo Grande 79070-900, Brazil; luapamaral@hotmail.com (L.A.d.A.); robertaserafim06@gmail.com (R.S.d.S.); elisvania@gmail.com (E.F.d.S.); ligiamacedo18@gmail.com (M.L.R.M.); 2Protein Purification Laboratory and its Biological Functions (LPPFB) of the Federal University of Mato Grosso do Sul, Campo Grande 79070-900, Brazil; anacristinaj@gmail.com; 3Faculty of Pharmaceutical Sciences, Food, and Nutrition, Federal University of Mato Grosso do Sul, Campo Grande 79070-900, Brazil

**Keywords:** nonalcoholic fatty liver disease, NAFLD, C57BL mice, obesity, high-fat diet

## Abstract

Researchers have a range of animal models in which to study Nonalcoholic fatty liver disease (NAFLD). Induction of NAFLD by a high-fat diet in the C57BL/6 strain is the most widely used among mice. In this study, we review works that performed NAFLD induction by a high-fat diet using the C57BL/6 strain, focusing on experiments on the effects of lipid ingestion. Studies are initially distinguished into researches in which mice received lipids by oral gavage and studies in which lipid was added to the diet, and each of these designs has peculiarities that must be considered. Oral gavage can be stressful for animals and needs trained handlers but allows accurate control of the dose administered. The addition of oils to the diet can prevent stress caused to mice by gavage, but possible changes in the consistency, taste, and smell of the diet should be considered. Regarding the experimental design, some variables, such as animal sex, treatment time, and diet-related variables, appear to have a definite pattern. However, no pattern was found regarding the number of animals per group, age at the beginning of the experiment, time of adaptation, the substance used as a vehicle, and substance used as a control.

## 1. Introduction

Nonalcoholic fatty liver disease (NAFLD) is the presence of hepatic steatosis at a level greater than or equal to 5% of the liver area without the presence of other diseases that affect the organ or the use of alcohol in quantities considered harmful [1]. The spectrum of the disease ranges from simple liver fat accumulation—so-called hepatic steatosis—to more severe progressions such as nonalcoholic steatohepatitis, cirrhosis, and hepatocellular carcinoma.

It is estimated that 25% of the world’s population has NAFLD [2]. In South America, this number reaches 30.45% [1]. Because some patients do not show changes in blood tests, the prevalence of the disease may be even higher [3].

The etiology of the disease is multifactorial, and its onset and progression depend on factors related to diet, weight, gut microbiota, in addition to the contribution of genetic factors [4,5,6]. The presence of the disease is correlated with obesity and is present in up to 91% of severely obese patients (Body mass index—BMI > 35 kg/m^2^) [7].

Obesity can reduce quality of life and is associated with unemployment and low productivity. According to the Global Health Observatory data repository, 39% of the world’s population over 18 were overweight (BMI ≥ 25), and 13.1% were obese (BMI ≥ 30) in 2016. Between 1996 and 2016, the number of obese people increased by 65.82% [8]. The situation is worsening in Latin America and the Caribbean, where about 58% of the population, i.e., 360 million people, are overweight, and the number of obese people reaches 23%, totaling 140 million people [9].

As no drug therapy for NAFLD treatment has been approved by the US Food and Drug Administration to date, researchers have been studying functional foods that can alleviate the disease [10]. Among them, lipids can contribute significantly to homeostasis recovery. Fatty acids, besides providing energy to the body, are precursors of signaling molecules and hormones and play an important role in gene regulation. When ingested, they can bind to specific receptors that change the functioning of nuclear transcription factors, and alter the expression of several genes depending on the composition of dietary lipids [11].

Researchers now have a range of animal models to research the effects of lipids on NAFLD. C57BL/6 mice, Wistar, and Sprague Dawley rats are generally the most used because of their intrinsic predilection for the development of obesity, type 2 diabetes mellitus, and NAFLD. NAFLD can be experimentally induced by animals with genetic changes (combined or not with hypercaloric diets), such as in leptin-deficient (OB/OB) mice models, which have an alteration in the gene responsible for leptin production, and leptin receptor deficient (DB/DB) mice, which have a mutation in the receptor coding gene for this same hormone [12].

There are models described as “chemical models”, in which the disease is caused by the administration of drugs such as streptozotocin or carbon tetrachloride. In addition to these, there are others that only involve dietary changes, such as a methionine and choline-deficient diet, a high fructose diet, and a high-fat diet [13,14,15].

Each model has its own characteristics and should be adopted according to the research objective. To help in this choice, there are several review studies in the literature that may help researchers [15,16,17,18].

In this study, we focus on NAFLD induction by a high-fat diet in C57BL/6 strain mice, which is the most commonly used strain for this experimental disease model [18]. High-fat diet intake allows animals to develop obesity, hyperinsulinemia, hyperglycemia, hypertension, and liver damage, similar to the phenotype observed in humans with NAFLD [12,19]. Thus, we review studies that use this model to study the effects of lipid intake. In addition, taking into account the articles found, as well as the experience of our group with this type of experiment, we focus on points not discussed in other reviews such as sex, age, and number of animals, the time of adaptation and treatment, amount and composition of fat diets, as well as dose, vehicle, and substances used as a control in the treatment. We also sought to present relevant information in this type of research, summarizing in tables and discussing possible barriers and doubts that may arise to the researcher who is considering conducting research using this model.

## 2. Materials and Methods

This review included articles published from 1 January 2014 to 13 May 2019, indexed in the National Center for Biotechnology Information Support Center’s PubMed. For the selection of articles, PubMed Advanced Search Builder was used using the terms: NAFLD or Nonalcoholic Fatty Liver Disease, C57BL or mice, and lipid.

We considered studies that performed experiments with C57BL/6 mice (non-knockout) with NAFLD induction by hypercaloric diet, and whose treatment was based on lipids.

The database search resulted in 1103 eligible articles. After reading the titles and abstracts, 34 articles were selected for a full reading. In the end, 15 articles met all selection criteria and became part of this review, as outlined in Appendix A.

First, we addressed the common general characteristics of both methodologies, such as the initial age of the animals, sex, number of mice per group, adaptation period, and treatment time. Then, due to the distinctions that lipid administration by gavage or diet implies, we chose to separate the studies into two distinct sections: oil administration included in diet and oil administration by oral gavage.

## 3. Results

In the fifteen studies that we analyzed, 46.67% performed the administration of oils by oral gavage, and 55.33% chose to insert the lipids in animal diets, as exemplified in Figure 1.

### 3.1. Experimental Design: General Aspects

Regarding the sex of the animals, all studies that presented this information claimed to have used male mice (Table 1). C57BL/6 male mice have been considered the standard in experiments with diet-induced obesity [20]. Yang et al. (2014) found out that male mice gain more body weight (93%, n = 269) than female mice (71%, n = 255), after 35 weeks of high-fat feeding. Furthermore, male mice fed on high fat show statistically significant differences in weight compared to the control than female mice [20].

No pattern was found regarding the number of animals, which ranged from a minimum of five to a maximum of 12 per group (Table 1). The age of the animals at the beginning of the experiment ranged from three to 15 weeks. However, some authors (20.00%) did not mention the age of the animals in the methodology. The adaptation time of the mice ranged from zero to two weeks.

To determine the existence and degree of hepatic steatosis, 60% of the studies used an imaging method (histology or stereology) combined with hepatic lipid quantification (using specific kits, colorimetric methods, or according to Folch et al. [21]). Regarding the others, 20% chose to perform only liver lipid quantification, and 20% opted for histological analysis alone.

The minimum treatment time was four weeks, and the maximum time was 36 weeks. However, 53.33% of the experiments adopted 12 weeks of treatment (Table 1). Obesity in C57BL/6 mice can be divided into three stages: the early stage, from week one to week four of high-fat feeding; the middle stage, from week four to week 15; and the late stage, from week 15 [37]. Therefore, the majority of studies opted to end the experiment during the late middle stage of obesity.

### 3.2. Oil Administration by Oral Gavage

#### 3.2.1. Diet

Regarding the amount of fat present in the diet of experiments that chose to provide the oil by gavage, there is a pattern regarding the number of calories from fat. All experiments analyzed opted for a diet with 60% of total kcal from fat (Table 2).

For the lean control groups, 57.14% of the experiments opted for a diet with 10% kcal from fat, 28.57% for 13.5% kcal from fat, and one study did not disclose the percentage of fat but mentioned using a standard rodent diet.

#### 3.2.2. Dose

After analyzing the studies that administered the oral oil via gavage treatment and the authors’ practical experience with this type of experiment, we surmise that this type of assay should be undertaken with an awareness of some important factors that differ from those inherent to other animal experiments.

Although gavage administration is the simplest method for performing accurate mice dosing, the stress caused by this procedure should be considered. In addition, there is a need for more time to perform daily procedures, and individuals responsible for administration should undergo specific training to ensure proper use of the technique, avoiding further damage to animals [38]. In an experimental study with C57BL/6 female mice (n = 20 per group), the possible adverse effects of using gavage with or without anesthesia were verified. The use of brief isoflurane as the anesthetic reduced the cases of incomplete intake of the content provided, and the loss of animals due to traumas caused by the technique. Thus, the authors advocate the use of anesthesia in studies that perform long-term gavage. Regarding weight gain, the control group that received no gavage and the groups that showed no differences in weight at the end of the experiment indicate that this technique may be a viable and unbiased alternative in experiments on obesity [39]. We reinforce that oral gavage can be a stressful procedure when performed improperly; researchers should be adequately trained to prevent possible animal deaths and excessive stress.

Table 3 shows the doses of oils administered to mice by gavage. Of the studies evaluated, 57.14% used values between 4 mg/kg/day and 125 mg/kg/day. In oil with a density of 0.8 g/mL, these values correspond to levels equivalent to 0.35–10.94 mL for a 70 kg person by direct conversion, or 0.028–0.7 mL if the conversion proposed by Reagan-shaw et al. is used [40].

In the study by Yu et al. (2017), 100 mg/kg of Omega-3 polyunsaturated fatty acids (PUFAs) were used, i.e., the amount of oil administered varied according to PUFA concentration. To circumvent this variation, the authors adjusted the final amount administered to the treatment groups using corn oil, and the control group received isovolumetric corn oil [23].

Another study, which defined the amount offered because of a specific component of the oil, was that of González-Mañán et al. (2017). The researchers administered 1.94 mg of alpha-linolenic acid (ALA) per gram of body weight of mice. According to the authors, the oil used had a concentration of 35% ALA, i.e., the final amount of oil offered was 5.54 mg of oil/g, equivalent to 5.54 g/kg. For mice with 30 g of body weight, this meant 166.2 mg of oil. Due to the low weight of a mouse, it can be difficult to understand how much this amount represents. To better assimilate this value, we can draw a parallel with a 70 kg person, to which 5.54 g/kg would represent 31.02 mL, according to the conversion proposed by Reagan-shaw et al. Thus, as we commented, caution should be taken when defining the quantity of oil administered.

Only one experiment adopted gavage once a week. The set values were 250 or 500 mg/kg of body weight, which represents a weekly amount of 7.5 or 15 mg for a 30 g mouse. Considering that the gavage was performed weekly, this amount was similar to that of most of the other studies analyzed.

#### 3.2.3. Vehicle

Depending on the amounts exemplified in the previous section and taking into account that oral gavage in mice is commonly performed with a proprietary gavage needle attached to a 1 mL insulin syringe, depending on the amount administered per animal, it is not possible to administer only the oil treatment and thus a vehicle substance is required. The choice of this vehicle may affect the study results, and it is essential that the authors describe in detail the procedure used. It is not possible to use an innocuous vehicle, such as a saline solution, because the oil treatment will not dissociate from it.

Only 28.57% of the studies made it clear to have used a vehicle in the methodology section. These studies mentioned having used olive oil or corn oil (Table 3). However, we emphasize that, according to the quantities administered, other experiments probably used vehicles and did not mention it in the methodology section of the studies.

#### 3.2.4. Control Substance

Another fundamental point that should be present in the methodology section of the studies is the indication of the substance offered to the control groups. As already mentioned, the possible stress created by oral gavage should be mimicked in the control groups. Another factor to be taken into consideration is the calories generated by the oil treatment, which should also be present in this group. Also, because it is a lipid experiment, the calories added by it should also be considered. If a control oil is chosen, it should be evaluated whether it has characteristics that may affect the results of the experiment, such as inflammatory or anti-inflammatory capacity, orexigenic or sacietogen potential, laxative factors, etc.

### 3.3. Oil Administration Included in the Diet

#### 3.3.1. Diet

The experiment design with the addition of oil to the diet does not allow exact control of the dose that each animal receives, only a percentage in relation to the total ingested in the diet. An advantage is that it avoids the stress caused by handling and storage, possibly not requiring daily handling of animals. However, it is important to emphasize that, according to the oil used, possible rancidity and oxidation of the compounds should be considered. To circumvent this adversity, researchers added 0.2 µg of t-butylhydroquinone/g of oil to mice diets and performed daily feed exchanges [30].

Unlike the experiments with gavage, we did not identify a pattern regarding the number of calories from fat in the experiments with added dietary oil. Quantities ranged from 32–60% of kcal from fat, as shown in Table 4. In the control groups, they ranged from 10 to 13%. Diets with 60% of the calories being from lipids are considered effective in promoting obesity and metabolic changes associated with the disease. However, some authors argue that this percentage exceeds the dietary habits common to Western diets [41,42].

The amount of sucrose in the diet of animals is also a factor that should be considered. Although both a high-fat diet and a high sucrose diet may lead to liver fat accumulation, this will happen through different mechanisms. For example, a sucrose-rich diet may lead to a further increase in de novo lipogenesis [43].

Regarding the groups with a hypercaloric diet, in 25% of the studies that we analyzed, there was no description of the amount of sucrose used. In 50% of the experiments, the amount was 100 g/kg. In the experiment that used the lowest amount, the diet consisted of 89 g/kg sucrose, but also 162 g/kg maltodextrin. Already the experiment that used more sucrose added a total of 203.75 g/kg.

It is noteworthy that the experiments that choose this design should adjust the percentage of kcal from fat in the control and treatment obese groups and even, depending on the objective of the work, consider balancing the amounts of saturated, monounsaturated and polyunsaturated fatty acids, for example, when comparing the ratio between omega-6 and omega-3. The authors highlighted the importance of matching the fat content (as shown in Table 5). Both experimental groups received equivalent amounts of saturated, monounsaturated, and polyunsaturated fatty acids, only differing in the ratio between omega-6 and omega-3. Thus, a possible bias caused by the difference between the percentage of fatty acid saturation was avoided. Therefore, the difference was limited to the type of polyunsaturated fatty acid in the diet [36].

In addition to the percentage change shown in Table 4, the composition of the diets also did not present a definite pattern (Table 5). Park et al. (2016) did not alter the sucrose content of the diet but did change the soybean oil of the control diet for the oil under study [30]. Bargut et al. (2014) maintained the amount of sucrose, increased the amount of casein in the obese groups in order to match the protein percentage with that of the control diet (10% of total protein calories). In addition, the lean treatment group received 36 g of fish oil in place of soybean oil, and the obese control group received 238 g/kg of fish oil while the obese control group received 238 g/kg of lard [31]. Chamma et al. (2017) Offered all groups the same amount of soybean oil and sucrose. Lean control groups received less casein and more corn starch, totalizing 75.8% of energy from carbohydrates, 14.3% of protein, and 9.9% of lipids. The obese groups received the same amount of casein, corn starch, and sucrose with different amounts of lard in change with medium-chain triacylglycerol (MCT) [32]. Beppu et al. (2017) also only change the amount of lard in the diet of the obese groups for starfish oil [33]. Soni et al. (2015) kept the same ingredients in all groups, the only change in diet was that the obese treatment group received 20 g/kg of fish oil and 30 g/kg of corn oil, while the obese control group received 50 g/kg of corn oil [34]. Yang et al. (2015) keep the same amount of all nutrients in the diet of experimental groups, with the exception of an exchange of 100 g/kg of lard for 100 g/kg of saury oil [35].

#### 3.3.2. Dose and Control Substance

The lipids used to compose the control diet and the animal treatment varied among experiments (Table 5). Park et al. (2016) conducted an experiment with four groups of C57BL/6 mice, two control groups with 10% of the energy from lipids, and two HFD groups with 45% of the energy from fat. One control group received all lipids from pine nut oil (PNO), and the other received all lipids from soybean oil (SBO). The high-fat diet (HFD) groups received 35% from lard and 10% from PNO or SBO [30]. The authors pointed out that SBO was chosen to compose the diet because its fatty acid composition is similar to that of PNO. To avoid oxidation of polyunsaturated oils, 0.2 µg of t-butylhydroquinone/g of oil was added to the diet.

However, some studies have changed the amount of lard in diets [33,35,44]. In the research conducted by Chamma et al. (2017), the fat control group received 236 g/kg of lard and 42 g/kg of soybean, and the treatment groups received 25%, 75% or 100% of medium-chain triacylglycerol (MCT) replacing lard, keeping the amount fixed at 42 g/kg of soy oil.

In 87.5% of the experiments analyzed, the authors chose to offer diets containing lard as the main source of lipids (Table 5). Of these, 57.14% made it clear that the diets of the treatment groups received the lard treatment oil, and only one experiment [30] chose to replace the soybean oil for the treatment oil of the control diet, maintaining the same amount of lard.

Only one experiment did not have a lard-based diet but rather used coconut oil. The authors of this research also opted to replace the treatment oil for the control diet’s corn oil [34].

The consistency, conservation, and palatability of the diet is a fundamental point in this type of experiment, as a change in any of these factors may bias the experiment. In this sense, Jurado-Ruiz et al. (2016) pointed out that the smell of fish oil may reduce the feed intake of animals. To circumvent this possible adversity, the authors added bacon flavoring to the diet, as well as 0.2 µg of t-butylhydroquinone/g oil, to avoid lipid oxidation [30].

## 4. Conclusions

We conclude by this review that there is standardization in the amount of fat percentage in the diet of gavage experiments, amount of dietary fat in control groups, type of fat used to compose most of the diet in diets with added oil, and the sex of the animals. However, we did not find a standardization regarding age and quantity of animals per group, time of adaptation, duration of the experiment, amount of lipids used in treatment, the vehicle used for storage, and the substance offered to control groups. It is worth mentioning that these are relevant aspects of the analysis of the quality and replicability of experiments. In some cases, they were not described in the methodology of the studies. Establishing a standard model can be a complicated task since defining the best methodology is practically impossible, given the many variables of biological models. However, standardization should be encouraged since its absence makes it difficult to advance researches in this field. Furthermore, it is essential that variables that may influence the final result to be described. Only then can new experiments confirm the results obtained by or that rely on the performance of new intervention proposals.

## Figures and Tables

**Figure 1 nutrients-11-03067-f001:**
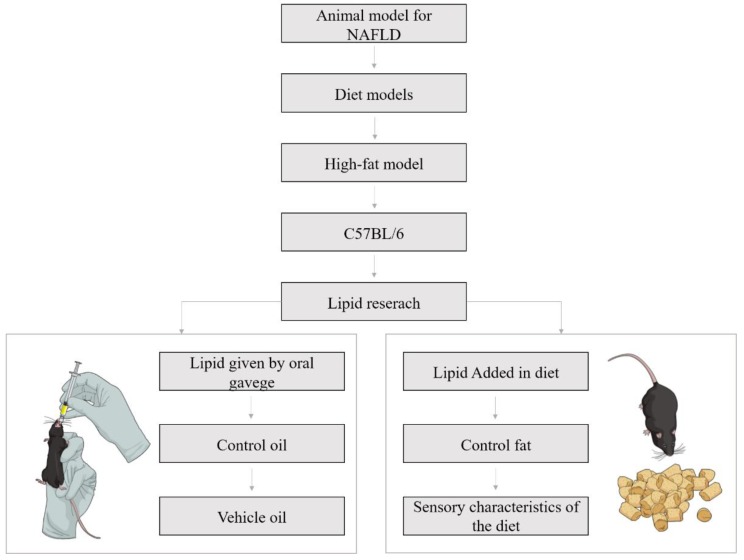
Nonalcoholic fatty liver disease in C57BL/6 models for lipid research: key points of experimental design.

**Table 1 nutrients-11-03067-t001:** Sex, number of animals per group, age at the start of the experiment, time of adaptation, and time of treatment of the animals.

Gavage					
Ref.	Gender	Number of Animals Per Group	Age of Animals	Weeks of Adaptation	Experiment Duration after Adaptation (Weeks)
[22]	Male	8	6-weeks-old	2	12
[23]	Male	12	3–4-weeks-old	1	16
[24]	Male	10	-	2	12
[25]	−	7 or 8	5-weeks-old	1	12
[26]	Male	5 in control groups	4-weeks-old	0	12
		9 in high-fat groups			
[27]	Male	10 or 11	−	−	12
[28]	−	5	−	−	4
**Lipids Included in Diet**					
[29]	Male	6	5-weeks-old	−	12 weeks to promote NAFLD + 24 weeks of treatment
[30]	Male	–	5-weeks-old	3 days	12 weeks
[31]	Male	10	3 months of age	−	8 weeks
[32]	Male	10	3 months of age	−	12 weeks
[33]	Male	−	4-weeks-old	7 days	8 weeks
[34]	Male	12	9-10-weeks-old	−	12 weeks
[35]	Male	10	5-weeks-old	2 weeks	18 weeks
[36]	Male	−	14-15-weeks-old	−	15 weeks

Note: “−“ means that the information was not provided by the authors. NAFLD, nonalcoholic fatty liver disease.

**Table 2 nutrients-11-03067-t002:** Group division and percentage of lipids in the diets of the experiments using gavage.

	Control Groups		Treatment Groups	
Ref.	Lean Control	Obese Control	Lean Treatment	Obese Treatment
[22]	13.5% kcal from fat	60% kcal fat content	−	60% kcal fat content
[23]	10% kcal from fat	60% kcal fat content	−	60% kcal fat content
[24]	13.5% kcal from fat	60% kcal fat content	−	60% kcal fat content
[25]	10% kcal from fat	60% kcal fat content	−	60% kcal fat content
[26]	10% kcal from fat	60% kcal fat content	10% kcal from fat	60% kcal fat content
[27]	10% kcal from fat	60% kcal fat content	10% kcal from fat	60% kcal fat content
[28]	standard mice diet	60% kcal fat content	−	60% kcal fat content

Lean control: Group that received a normocaloric diet and a control substance by oral gavage. Obese control: Group that received a high-fat diet and a control substance by oral gavage. Lean treatment: Group that received normocaloric diet and the studied lipid by oral gavage. Obese treatment: Group that received a high-fat diet and the studied lipid by oral gavage.

**Table 3 nutrients-11-03067-t003:** Dose and frequency of oil supplemented by oral gavage, gavage vehicle, and control substance.

Ref.	Dosage of Oil Given by Oral Gavage	Gavage Vehicle	Substance Administered to Control Groups
[22]	12.5, 62.5 or 125 mg/kg/day	−	Olive oil
[23]	100 mg/kg/day of Omega-3 PUFAS from fish oil, MO low DHA purity or MO high DHA purity	Corn oil	Corn oil
[24]	25, 50 or 100 mg/kg/day	Olive oil	Olive oil
[25]	250 or 500 mg/kg/week	−	Vehicle
[26]	1.94 mg ALA/g animal body weight/day	−	Saline solution
[27]	50 mg/kg/day	−	Saline solution
[28]	4, 8 or 16 mg/kg/day	−	−

Note: PUFAs, polyunsaturated fatty acids; MO, microalgal oil; DHA, docosahexaenoic acid; ALA, alpha-linolenic acid.

**Table 4 nutrients-11-03067-t004:** Group division and percentage of lipids in the diets of the experiments using lipids in studies that added it to the diet.

	Control Group		Treatment Group	
Ref.	Lean Control	Obese Control	Lean Treatment	Obese Treatment
[29]	13% kcal from fat	49% kcal fat content	Not applicable	49% kcal fat content
[30]	10% kcal from fat	35% kcal fat content	10% kcal from fat	35% kcal fat content
[31]	10% kcal from fat	50% kcal fat content	10% kcal from fat	50% kcal fat content
[32]	10% kcal from fat	50% kcal fat content	Not applicable	50% kcal fat content
[33]	−	46% kcal fat content	Not applicable	46% kcal fat content
[34]	12% kcal from fat	32% kcal fat content	Not applicable	32% kcal fat content
[35]	−	60% kcal fat content	Not applicable	60% kcal fat content
[36]	−	46,3% kcal fat content	Not applicable	46,3% kcal fat content

Lean control: Group that received a normocaloric diet. Obese control: Group that received a high-fat diet. Lean treatment: Group that received a normocaloric diet and the studied lipid replacing some component of the diet. Obese treatment: Group that received a high-fat diet and the studied lipid replacing some component of the diet.

**Table 5 nutrients-11-03067-t005:** Composition of the diets of the experiments using lipids in studies that added it to the diet.

Ref.	Diet Composition
[29]	HFD Control: Lard-based diet 49% total kcal of diet from fat
	HFD Treatment: 41.7% total kcal of diet from treatment oils
[30]	Lean Control: 45 g/1045.0 g (10% of kcal) from soybean oil and 350 g/1045.0 g of sucrose
	Lean Treatment: 45 g/1045.0 g (10% of kcal) from pine nut oil and 350 g/1045.0 g of sucrose
	HFD Control: 157.5 g/848.1 g of lard (35% of kcal), 45 g/848.1 g (10% of kcal) from soybean oil and 172.8 g/848.1 g of sucrose
	HFD Treatment: 157.5 g/848.1 g of lard (35% of kcal), 45 g/848.1 g (10% of kcal) from pine nut oil and 172.8 g/848.1 g of sucrose
[31]	Lean Control: 40 g/kg of soybean oil, 100 g/kg of sucrose and 140 g/kg of casein
	Lean Treatment: 36 g/kg of fish oil, 4 g/kg of soybean oil, 100 g/kg of sucrose and 140 g/kg of casein
	HFD Control: 238 g/kg of lard, 40 g/kg of soybean oil, 100 g/kg of sucrose and 175 g/kg of casein
	HFD Treatment: 238 g/kg of fish oil, 40 g/kg of soybean oil, 100 g/kg of sucrose and 175 g/kg of casein
[32]	Lean Control: 42 g/kg of soybean oil, 622.69 g/kg of cornstarch, 100 g/kg of sucrose, 42 g/kg of soybean oil and 136 g/kg of casein
	HFD Control: 236 g/kg of lard, 42 g/kg of soybean oil, 352.192 g/kg of cornstarch, 100 g/kg of sucrose, 42 g/kg of soybean oil and 170 g/kg of casein
	HFD + 25% MCT: 177 g/kg of lard, 42 g/kg of soybean oil, 61.5 g/kg of medium-chain triacylglycerol, 350.192 g/kg of cornstarch, 100 g/kg of sucrose, and 170 g/kg of casein
	HFD + 75% MCT: 59 g/kg of lard, 42 g/kg of soybean oil, 184,5 g/kg of medium-chain triacylglycerol, 345.192 g/kg of cornstarch, 100 g/kg of sucrose, and 170 g/kg of casein
	HFD + 100% MCT: 42 g/kg of soybean oil, 246 g/kg of medium-chain triacylglycerol, 342.692 g/kg of cornstarch, 100 g/kg of sucrose, and 170 g/kg of casein
[33]	HFD Control: 200 g/kg of lard, 200 g/kg of casein, 250 g/kg of corn starch, 100 g/kg of fructose, 100 g/kg of sucrose, and 50 g/kg of soybean oil
	HFD + 2% of Starfish Oil: 180 g/kg of lard, 200 g/kg of casein, 250 g/kg of corn starch, 100 g/kg of fructose, 100 g/kg of sucrose, 20 g/kg of starfish oil and 50 g/kg of soybean oil
	HFD + 5% of Starfish Oil: 150 g/kg of lard, 200 g/kg of casein, 250 g/kg of corn starch, 100 g/kg of fructose, 100 g/kg of sucrose, 50 g/kg of starfish and 50 g/kg of soybean oil
[34]	Lean Control: 222 g/kg of casein, 50 g/kg of sucrose, 560 g/kg of corn starch, 50 g/kg of cellulose, 25 g/kg of corn oil and 25 g/kg of coconut oil
	HFD Corn Oil Control: 256 g/kg of casein, 100 g/kg of sucrose, 348 g/kg of corn starch, 58 g/kg of cellulose, 100 g/kg of coconut oil and 50 g/kg of corn oil
	HFD Treatments: 256 g/kg of casein, 100 g/kg of sucrose, 348 g/kg of corn starch, 58 g/kg of cellulose, 100 g/kg of coconut oil, 30 g/kg of corn oil + 20 g/100 g of EPA and DHA-enriched oils
[35]	HFD Control: 258 g/kg of casein, 162 g/kg of maltodextrin, 89 g/kg of sucrose, 65 g/kg of cellulose, 320 g/kg of lard, and 32 g/kg of soybean oil
	HFD Treatment: 258 g/kg of casein, 162 g/kg of maltodextrin, 89 g/kg of sucrose, 65 g/kg of cellulose, 220 g/kg of lard, 32 g/kg of soybean oil and 100 g/kg of saury oil
[36]	HFD Control: 40.8% SFA, 42.0% MUFA and 17.3% PUFA (15.9% Omega-6 and 1.4% Omega-3) (Lard and soybean-based diet)
	HFD Treatment: 41.5% SFA, 41.4% MUFA and 17.1% PUFA (12.5% Omega-6 and 4.6% Omega-3) (Lard, soybean oil and Menhaden fish oil-based diet)

Note: HFD, high-fat diet; MCT, medium-chain triacylglycerol; DHA, docosahexaenoic acid; EPA, eicosapentaenoic acid; MUFA, monounsaturated fatty acids; PUFAs, polyunsaturated fatty acids. Lean control: Group that received a normocaloric diet. HFD control: Group that received a high-fat diet. Lean treatment: Group that received a normocaloric diet and the studied lipid replacing some component of the diet. HFD treatment: Group that received a high-fat diet and the studied lipid replacing some component of the diet.

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
