# Peer review of "Nonalcoholic Fatty Liver Disease Induced by High-Fat Diet in C57bl/6 Models"

_nutrients, 2019, doi:10.3390/nu11123067_

Round 1
Reviewer 1 Report
Introduction section: NAFLD is actually the main cause of chronic liver disease worldwide. Its pathogenetic mechanisms involve are due to a metabolic profile expressed within the context of a genetic predisposition and is associated with a higher energy intake. NAFLD patients have more than one feature of Metabolic syndrome. In fact NAFLD has been considered the hepatic components of the Metabolic syndrome (PMID:27610012). Also it is plausible that genetic factors could play a pivotal role in the pathogenesis of NAFLD. The recent advances in genomics, transcriptomics, and proteomics have highlighted new pathogenic pathways. Increasing literature data support the role of single nucleotide polymorphisms, and in particular of genes involved in insulin signaling, lipid homeostasis, and oxidative stress (PMID:31098374) Methods section: i suggest to improve the description of the diets used in the included studies, and in particular the daily meals. Results section: please describe the tool reported in the studies to detect hepatic fat accumulation Discussion section: i sugget to Author, to emphasize the limits of the described studies, as a vehicle not reported, and to highlight the role of the genetic pattern in the NAFLD development and progression.
Reviewer 2 Report
In this review article, the authors discuss the most widely used experimental model of NAFLD - high-fat-diet (HFD) inC57BL/6 strain mice. They summarize in tables different variation of HFD and discuss many critical factors such as age, gender, number of animals, appropriate controls. The subject of the review is interesting and can be very useful for future experimental research. Still I have comments/questions:
The abstract should be changed. It should be more condense and concentrated and better represent the entire paper. Fig 1 should be moved to supplementary materials Line 116: “Obesity in C57BL/6 mice can be divided into three stages” but here we talk not only about obesity. The different stages of NAFLD should be mentioned. Table2, 4, 5. The explanation for the following groups is requiered: the obese control/obese treatment, lean control/lean treatment. In my opinion the oral gavage model is not widely used anymore. Line 95, in how many papers the gavage was used (% vs diet)? Can daily gavage be classified as mild procedure? The classification criteria that underlie this valuation have been established by the Expert Working Group on severity classification of scientific procedures per-formed on animals and can be found at http://ec.europa.eu/environment/chemicals/lab_animals/pdf/report_ewg.pdf. Based on our experience,oral administration is very stressful for animals and more challenging from the technical point of view, requiring a certain level of expertise from the lab personnel performing the tests. It frequently introduces inadvertent tracheal administration, aspiration pneumonia, oesophageal perforation and even gastric rupture, resulting in the removal of animals from the study. The authors should emphasize this point in the review. The diet part should be extended. Can authors comment on western diet and the cholesterol level? Two principal difference between chow and defined diets are the phytoestrogen content from soy and sucrose. Could the authors clarify this point
Round 2
Reviewer 2 Report
The authors have done an excellent job in addressing the issues raised during original review
Author Response
We were happy to have answered the suggestions. And thanks again to the reviewer.